# Cognitive Ergonomics Evaluation Assisted by an Intelligent Emotion Recognition Technique

**Adrian Rodriguez Aguiñaga \***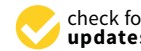**, Arturo Realyvazquez**, **Miguel Angel Lopez Ramirez and Angeles Quezada**

Computer Systems and Industrial Department, Tecnológico Nacional de México/IT de Tijuana, Tijuana 22430, Mexico; arturo.realyvazquez@tectijuana.edu.mx (A.R.); mlopez@tectijuana.edu.mx (M.A.L.R.); mangelesquezada@gmail.com (A.Q.)

\* Correspondence: adrian.rodriguez@tectijuana.edu.mx; Tel.: +52-664-529-9998

**Featured Application: Currently, in Mexico, a regulation for the cognitive care of workers entered into force. This regulation is focused on reducing the conditions that could affect the mental health of a worker, and this work could be beneficial for compliance with these regulations.**

**Abstract:** The study of the cognitive effects caused by work activities are vital to ensure the well-being of a worker, and this work presents a strategy to analyze these effects while they are carrying out their activities. Our proposal is based on the implementation of pattern recognition techniques to identify emotions in facial expressions and correlate them to a proposed situation awareness model that measures the levels of comfort and mental stability of a worker and proposes corrective actions. We present the experimental results that could not be collected through traditional techniques since we carry out a continuous and uninterrupted assessment of the cognitive situation of a worker.

**Keywords:** awareness; emotions; ergonomic; cognitive; machine learning

## 1. Introduction

There is no doubt that the technological deployment of the last decades has considerably increased the activities that a person can perform in a company; however, technology also contributes to the generation of new problems such as information overload or affectation in short- and long-term memory. According to the International Ergonomics Association (IEA), ergonomics considers the cognitive processes that are concerned with mental processes (cognitive ergonomics), such as perception, memory, reasoning, and motor response, and how they affect the interactions among humans and systems [1]. Cognitive ergonomics emphasizes the analysis of mental functions and the design of user-centered systems that support cognitive tasks processes, to reduce the psychological wear [2]. Besides that, several studies have been carried out to address cognitive problems [3–5], most of them implementing traditional techniques for data collection, such as surveys or interviews. Although some proposals consider the implementation of technological strategies in data collection, they are invasive and could affect the experimental conditions.

This work proposes a non-invasive solution by implementing an intelligent pattern recognition algorithm through video analysis or facial emotion recognition (FER), which has had significant advances in the area in the last decade, mainly to the democratization of machine learning and advances in GPU technologies. Particularly, deep learning has shown competitive results in the FER analysis, and it is the convolutional neural networks (CNN) that have achieved the best results [6,7]. The premise is that it is possible to establish a relationship between the emotional states detected in the facial expressions of the worker to identify critical routes that may present opportunities to reduce

human error, e.g., identifying external factors, such as personal problems or discomfort caused by the company's environment. Although several other techniques can be implemented to study human emotions, such as electroencephalogram, heart rate analysis, and electromyography [8–13], these are invasive and, in some cases, expensive to implement, and their technical feasibility makes it impossible to apply them in noisy environments. This is the reason why we decided to implement a technique based on video analysis, despite the problems associated with it such as noise and data variability.

This work studies the relationship between the employees and their environment by considering that the employee perceives a wide range of stimuli, and we expect that the recognition process will help to identify and thus prevent cognitive problems.

## 2. Materials and Methods

### 2.1. Affective Computing

Affective computing studies and develops systems or devices that can recognize, interpret, process, or simulate human emotions to achieve a more natural interaction between humans and machines. It is an interdisciplinary field spanning computer science, psychology, and cognitive science [14] that focuses on the sense of the emotional states of users and performs specific actions according to them. Affective computing considers emotions as fundamental to the human experience, but technologists have ignored it so far. However, the more computers we have in our lives, the more we are going to want them to be socially smart.

#### 2.1.1. Computational Emotions Model

Emotions are a very difficult subject to deal with since there is no clear definition of what is an emotion. Nevertheless, this work implements the Ekman, Russell, and Scherer models to correlate emotions with physiological responses and establish the classes that propose them as measurable situations [15–19]. The Russell model provides arousal and valence combinations as high arousal and high valence (HA/HV, happiness), high arousal and low valence (HA/LV, anger), low arousal and low valence (LA/LV, sad) and, finally, low arousal and high valence (LA/HV, relaxed) [15]. We obtained the tags or names of the classes based on Ekman's primary distribution of emotions [18] (anger, disgust, fear, happiness, sadness, surprise), and from the Scherer model, we understand that emotions cause physiological responses that can be measured [19].

#### 2.1.2. Facial Expression Recognition

According to Ekman and Friesen [17], emotions can be expressed as microexpressions or subtle expressions. Microexpressions are involuntary facial movements that cannot be created intentionally and usually do not exhibit the characteristics of spontaneous occurrence, and subtle expressions occur when a person's response to an emotional situation is of low intensity or when a person is just starting to feel an emotion [20]. People use facial expressions to understand and express nonverbal messages, and we implement a pattern recognition algorithm to identify the gestural combinations, correlate them to an Endsley's level, and give recommendations based on the cognitive analysis of workers in real-time.

Automatic Facial Expression Recognition

Facial expression recognition plays an important role in various fields in recent decades; many practical applications have been found, such as human–machine interaction, affective computing, surveillance, and robot control [21]. Most of the developed approaches focus on representing variations of appearances of different facial features (muscle motions) to interpret the range of individual expressions as one feature and look for a pattern among them to recognize the different expressions. The implementation of these algorithms requires very large and varied data due to the fact that they operate in highly restricted dimensions or conditions such as aging factors, spontaneous expressions,

craniofacial growth, and different pose and noise variations, as well as imaging and lighting conditions; they are also limited by the context in which the data are collected. Hence, an unambiguous feature descriptor is a key component of expression recognition. However, constructing a stable descriptor, which is robust against such changes, is a challenging task [21].

### 2.1.3. Artificial Neural Networks

Artificial neural networks (ANN) have become a very relevant technique for classification and pattern recognition. ANN is a relatively competitive and versatile machine learning (ML) technique that incorporates functions or methodologies in their architecture to attend specific needs or situations of the problem. Traditionally, to work with images and video, we work with convolutional neural networks (CNN), and although there have been good results working with this type of architecture, it has recently been observed that Extreme Sparse Learning (ESL) improves the performance in scenarios where CNNs have had problems. Furthermore, traditionally, facial emotion recognition systems have been evaluated in highly controlled data, which are not representative of the environment faced by real-world applications. The proposal of the authors of [22] can jointly learn a dictionary and a nonlinear classification model, to allow accurate classification of noisy signals and imperfect data recorded in natural environments [22,23]. Figure 1 shows the architecture and the optimizer implemented in this work, and it can be observed that the architecture is based on a convolutional model and is optimized by the ESL algorithm.

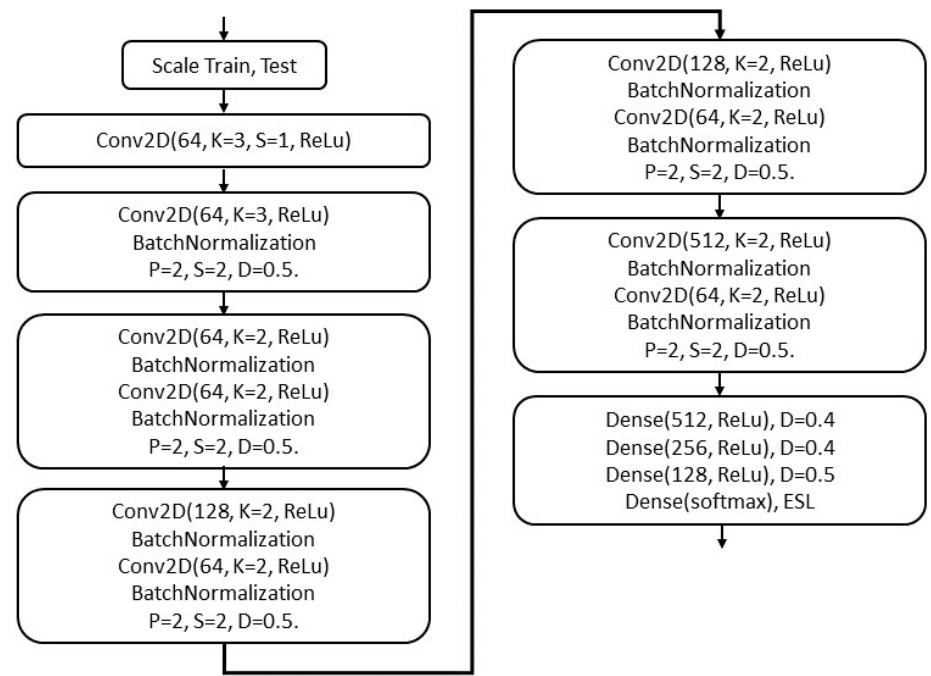

**Figure 1.** Neural network architecture implemented for automatic facial expression recognition.

### 2.2. Cognitive Ergonomics

Cognitive ergonomics studies the mental processes of a worker and how they affect the interactions among humans and systems; it also emphasizes the analysis of mental processes by designing user-centered systems to support the cognitive understanding of work situations and the limitations of human cognition [24]. Cognitive ergonomics ensure good usability of systems, by including several relevant topics such as mental workload, decision-making, skilled performance, human-computer interaction, human reliability, work stress, training, and how a product matches the cognitive capabilities of the human perception, mental processing, and memory [25,26].

Situation awareness

According to the authors of [27–29], situation awareness (SA) is the knowledge and understanding of specific situations or environments. SA is described as the perception of the elements in the environment within a volume of time and space, the comprehension of their meaning, and the projection of their status in the near future. We propose a SA model based on Endsley's model [30] that can be observed in Table 1.

**Table 1.** Endsley's model of situation awareness (SA) developed for this work. According to the authors of [30–33], creating a proper environment to motivate employees will increase their satisfaction, performance, and productivity.

| Emotion | Level 1: Recognition of Element | Level 2: Definition of Situation | Level 3: Projected Situation | Decision/Recommended Actions |
|---|---|---|---|---|
| Happiness | • Mental workload (stress)<br>• Variety/monotony of tasks<br>• Social relationships<br>• <br>• Participation/skills development<br>• Motivation<br>• Personal goals and objectives | • Mental workload is acceptable; the employee has no stress; he/she is happy and motivated with the tasks he/she performs, and enjoys the relationships with other employees. | • The employee will stay in the company for a long time and will improve his/her performance.<br>• The probability of social problems with other workers is very low. | • No actions are required in this specific task |
| Anger | • Mental workload (stress)<br>• Variety/monotony of tasks<br>• Social relationships<br>• <br>• Participation/skills development<br>• Motivation<br>• Personal goals and objectives. | • At least one of the elements in the environment causes anger in the employee. He/she does not enjoy all his/her work in the current environment | • The employee will not keep working in the company for a long time, and he/she can worsen his/her performance.<br>• Social problems with other employees can arise. | • Promote the acquisition and development of new skills through a variety of tasks.<br>• Promote teamwork.<br>• Recognize the employee´s effort.<br>• Take employee´s ideas into account.<br>• Decrease the mental workload. |
| Neutral | • Mental workload (stress)<br>• Variety/monotony of tasks<br>• Social relationships<br>• <br>• Participation/skills development<br>• Motivation<br>• Personal goals and objectives. | • Work does not cause happiness or anger to the worker. | • He/she only works with necessity and family obligation.<br>• It is impossible to define the time he/she will keep working in the company. | • Implement actions as in the anger case to change the employee´s emotions to happiness. |
| Sadness | • Mental workload (stress)<br>• Variety/monotony of tasks<br>• Social relationships<br>• Participation/skills development<br>• Motivation<br>• Personal goals and objectives | • At least one of the elements in the environment causes sadness in the employee.<br>• He/she does not enjoy all his/her work in the current environment. | • The employee will not keep working in the company for a long time, and he/she can worsen his/her performance.<br>• Employee´s motivation will decrease if the current environment remains as it is. | • Implement actions as in the anger case to change the employee´s emotion to happiness. |

Table 1 shows the actions to be taken before the various scenarios that can be detected with the platform. The recommendations and actions are based on Endley's model and are prepared by an expert in ergonomics. This proposal seeks to evaluate, model, and propose actions based on recognizing the environment, defining and projecting the situation, and issuing a decision and recommendations for each situation of the work environment, as:

- Perception of the elements in the environment. The manager/ergonomist identifies the factors and their current status that impact on the employee's emotions. These elements can be mental workload (stress), variety/monotony of tasks, social relationships, participation/skills development, motivation, personal goals, and objectives.

- Comprehension of the current situation. The manager/ergonomist understands if some elements are currently in an optimal status or if they require improvements.
- Projection of the current status. The manager/ergonomist projects a future situation of the employee's permanence and performance and the actions required to change the employee's emotions to happiness.

### 2.3. Experimental Setup

#### 2.3.1. Pattern Recognition Considerations

The recognition and classification process implements an algorithm (ESL), to recognize facial emotions in real-world natural situations [22]. This approach combines the discriminating power of Extreme Learning Machine (ELM), the reconstruction capability of sparse representation, and the inherent ability to reconstruct the original signals from noisy and imperfect samples based on a learned dictionary [22]. The assumption is that the underlying sparse representation of the natural signals or images is efficiently approximated by the linear combination of fewer elements of a dictionary. Furthermore, the dictionary can be obtained by applying predefined transforms to the data or from direct learning of training data. Since it usually leads to a satisfactory reconstruction, the function is defined as:

$$\min(X, D)\left(\| Y - D \|_2^2\right), \| x_i \| \le N_0. \tag{1}$$

Letting $Y$ be a set of input signals of dimension $N$. Learning a re-constructive dictionary for sparse representation of Y can be accomplished by solving $D = [d_1 d_2 \cdots d_M] \in \Re^{NxM}$, that is the learned over-complete dictionary, $X = [x_1 x_2 \cdots d_M] \in \Re^{MxS}$ the sparse code matrix of the input signals, $N_0$ is the sparsity constraint, and $\| \cdot \|$ is the pseudo norm that counts the number of non-zero elements.

Extreme Learning Machine (ELM) is a very competitive classification technique, especially for multi-class classification problems. It also requires less optimization, which results in a simple implementation, fast learning, and better generalization performance. The objective function of ELM can be summarized as proposed by the authors of [22], as shown in Equation (2):

$$\min(\beta)\left(\| H(X)\beta - Z \|_2^2 + \| \beta \|_2^2\right), \tag{2}$$

where $X$ denotes the set of training samples, $H$ is the hidden layer output matrix ($H(X) \in \Re_S^L$, $L$ is the number of nodes in the hidden layer, $\beta$ is the output weight vector of length $L$, and $Z$ is the vector of class labels of length $S$.

#### 2.3.2. Pattern Recognition for the Emotions

Only four of the emotional states of Ekman's model were chosen to carry out this experiment, (anger, happiness, sadness, and neutral) since this delimitation provides greater control of the classes when they belong to the same family, e.g., joy, satisfaction, and happiness belong to the same class due their relationship in the Russell model.

- Arousal: ranges from inactive (e.g., uninterested, bored) to active (e.g., alert, excited).
- Valence: ranges from unpleasant (e.g., sad, stressed) to pleasant (e.g., happy, elated).

#### 2.3.3. Datasets

The implemented pattern recognition algorithm identifies small changes in the facial expression as in [34], and we apply the CASME, CASME II, and fer2013 database to validate the training process, the details of the databases can be seen in Table 2. CASME is a facial expression collection project, containing information on 19 study subjects and 195 different samples related to basic emotions, tension, and repression. CASME II [35] is an extension of CASME that includes three other databases, USF-HD [36], Polikovsky's [37], and SMIC [38], thus adding a total of 26 study subjects to the experiment and

extending the model to define negative and positive emotions. Lastly, fer2013 is an open-source dataset which consists of 35,887 grayscale, 48 × 48 sized face images with various (seven) emotions, all labeled [39].

**Table 2.** Dataset descriptions and metadata composition.

| Dataset | Examples | Microexpressions | Frames |
|---------|----------|------------------|--------|
| CASME | 452 | 4 | 720 × 1280/480 × 640 |
| CASME II | 3000 | 5 | 280 × 340 |
| Fer2013 | 28,709 | 7 | 48 × 48 |

## 2.4. Experimental Methodology

Due to our proposal implementing intelligent computing techniques to perform constant evaluations and establish criteria for cognitive ergonomic action, we present an experimental exercise (see Figure 2) that shows the viability of implementation with an active diagnosis of the workers that can obtain constant, reliable, and automated information.

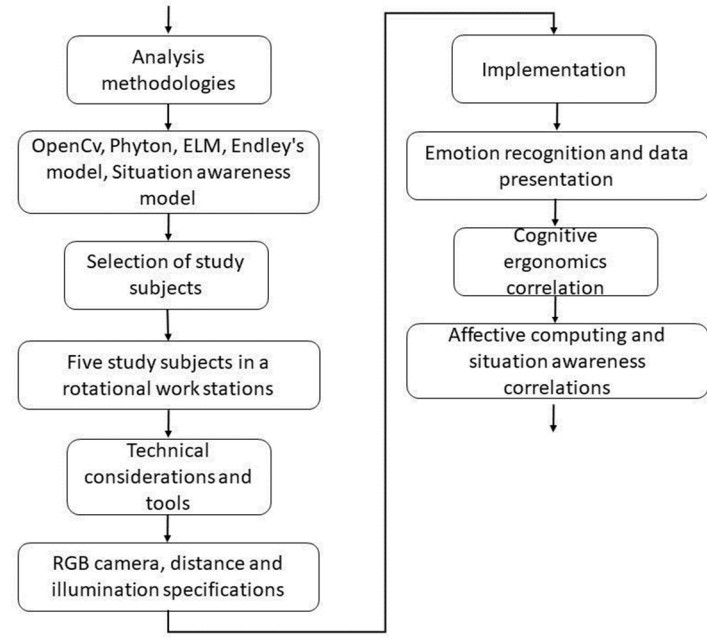

**Figure 2.** Activities and expected results of the stages of implementation of computer and ergonomic techniques.

### 2.4.1. Analysis Methodologies

Our models were implemented in Python, using OpenCV haarfaces detection and the Extreme Learning Machine algorithm described in the previous subsection, due to the technical capabilities and robustness of these platforms. Furthermore, our SA model was developed (see Table 1) to correlate emotional states and cognitive ergonomics.

### 2.4.2. Selection of Study Subjects

The company we work with specializes in the work of composite materials; most of its activities are handmade, so doing these activities requires skill and concentration. We select only one work station since it is the only one in which information is obtained from various workers without moving the camera (due to the personnel rotating every half hour), and it is the only one that involves both

genders in the process. The participants were in an age range between 24 and 37 years old (three men and two women) in the manufacturing area, all of whom work standing.

### 2.4.3. Technical Considerations and Tools

We used a Creative Senz3D™ (BasterX- Intel Real Sense technology. US manufactured) webcam with three lenses to capture visual data. However, for this experiment, we only used the RGB camera with a resolution of $1280 \times 720$, with a minimum illumination of 1000 lux. Figure 3 shows that the employee was not regularly looking towards the location where the camera was mounted; however, the webcam will be able to search and register the face. The platform was continuously executed for two weeks (the work schedule of the company is from 7:00 am to 5:00 pm, Monday to Saturday).

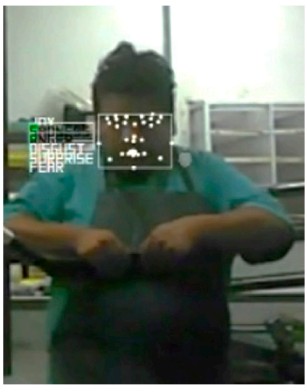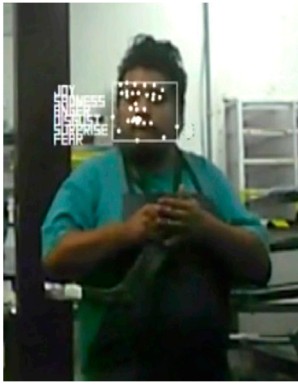

**Figure 3.** The employees were located within a range of three meters from the camera (0.3 yds) to perform active face tracking and continuously monitor their emotional states as can be observed.

### 2.4.4. Implementation

The data of the experiments are displayed in full-screen mode in bar graphs, as can be seen in Figure 4. Presenting the information in this way means that the ergonomics can quickly observe the behavior trends in their workspaces. The information is presented soberly because it is not intended that the system attracts too much attention or affects workers in any way. This information is continuously saved in a .csv file that can be accessed on demand.

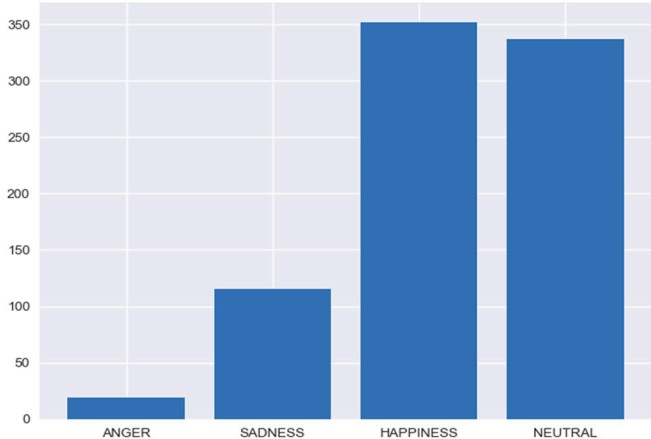

**Figure 4.** Data displayed to the ergonomist; each bar represents the number of times an emotional event has been recorded, and the figure is updated every 5 seconds.

Figure 4 shows that an ergonomist can easily observe which emotion or emotions are the most present states in the work environment and establish a relation between them and the SA of the

employees by consulting the proposed Endley's model to create a referential action to correct or maintain the situation of the work environment.

## 3. Results

### 3.1. Recognition Results

Since this work implements a technological development that supports an ergonomics activity, the first step is to analyze the stability of the application. Figure 5 shows an implementation of the detector, in which you can observe the placement of the landmarks and a comparison between two emotional situations. The experimental setup achieves up to 88% of the overall recognition rate in multiclass detection (see Figure 6).

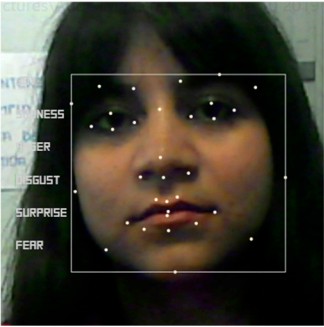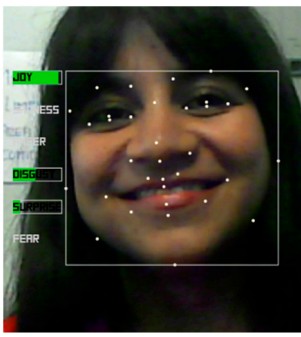

**Figure 5.** The orientation of the reference points for the detection of moods. The displacement of these points is computed to identify expressions. Figure 6 illustrates the mean classification rates and the observed standard deviation values obtained by applying S-ELM formulation. Note that for some of the databases, the individual performance was lower; however, the detector implemented was trained with the general integration of the databases. This means that in very noisy scenarios, we have recognition rates of 68% and for controlled scenarios up to 96% in the recognition rate, but it is regularly stable at 88%.

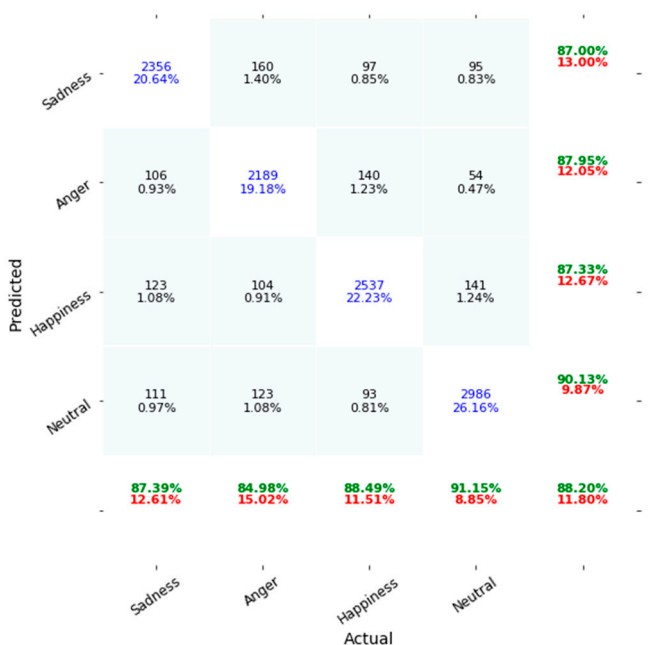

**Figure 6.** Accuracy of the training and validation for the multi-class classification and recognition application.

### 3.2. Experimental Results

Tables 3 and 4 show a similar number of events related to sadness and happiness, and Figures 7 and 8 present the overall institutional health information due to continuous monitoring. The system can only provide information in general and not in particular, so we cannot say which of the employees experience the emotions (due to the company privacy concerns).

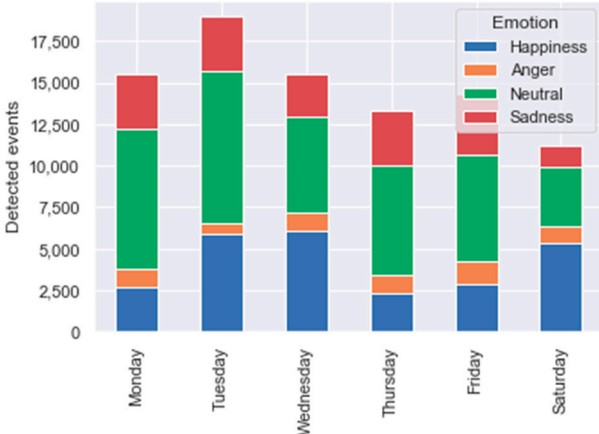

**Figure 7.** Emotion identification of the HA/HV, HA/LV, LA/HV, LA/LV states week 1.

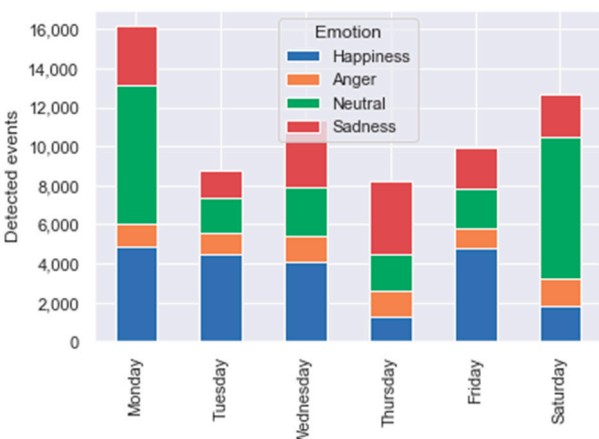

**Figure 8.** Emotion identification of the high arousal and high valence (HA/HV), high arousal and low valence (HA/LV), low arousal and low valence (LA/LV), and low arousal and high valence (LA/LV) states week 2.

**Table 3.** Overall results per day and per week; results by emotion to the day and the week (week 1).

| Emotion | Monday | Tuesday | Wednesday | Thursday | Friday | Saturday | Week |
|---|---|---|---|---|---|---|---|
| **Happiness** | 2697 | 5919 | 6057 | 2311 | 2856 | 5365 | 25,205 |
| **Anger** | 1043 | 582 | 1109 | 1129 | 1414 | 1024 | 6301 |
| **Neutral** | 8466 | 9178 | 5819 | 6549 | 6385 | 3563 | 39,960 |
| **Sadness** | 3282 | 3287 | 2482 | 3294 | 3650 | 1264 | 17,259 |

**Table 4.** Overall results per day and per week; results by emotion to the day and the week (week 2).

| Emotion | Monday | Tuesday | Wednesday | Thursday | Friday | Saturday | Week |
|---|---|---|---|---|---|---|---|
| **Happiness** | 4843 | 4437 | 4085 | 1304 | 4772 | 1793 | 21,234 |
| **Anger** | 1226 | 1106 | 1287 | 1328 | 1040 | 1462 | 7449 |
| **Neutral** | 7072 | 1851 | 2499 | 1807 | 1999 | 7259 | 22,487 |
| **Sadness** | 3017 | 1359 | 3490 | 3760 | 2122 | 2181 | 15,929 |

## 4. Discussion

Cognitive ergonomics is focused on the prevention of "human failures," e.g., when an accident occurs and expert studies fail to detect any mechanical failure or physiological cause which can attribute responsibility; usually, the conclusion is that the accident was due to "human error" (a person in perfect health has made an incomprehensible error) and it is at this point only when the traditional methodologies begin to analyze other factors, such as the emotional processes, even though emotions are one of the primary mechanisms in human behavior.

Our goal is the development of a system that a priori could help to mitigate the effects of situations that can affect the employee's environment, focused on conducting continuous studies that allow early attention to conditions that could affect organizational and personal health, and the possibility of using intelligent computation to support problems that due to their nature are hard to solve. However, the challenge is that the design of the scenarios that model this kind of question requires a multidisciplinary effort, and many of the relationships between them are not yet defined. Nevertheless, our work takes advantage of many already developed tools and exploits the relationships of ergonomics and affective computing, allowing the implementation of analysis that could be hardly approached by traditional perspectives such as carrying out exhaustive studies, uninterrupted processes and other technical or human limitations, but understanding that human intuition cannot be substituted. Furthermore, some proposals that interrelate aspects of the affective computing and cognitive ergonomics interaction are presented, e.g., it was beneficial that the relationship between the Ekman and Russell models match with the Endsley's model for the SA considerations to define the proposed scenario as a cognitive ergonomics problem.

As already mentioned, although this work presents a solution based on pattern recognition techniques, it is difficult to establish parameters that can be generalized, e.g., some of the experimental conditions were chosen to fit the needs of the experiment. The first thing that we need to explain is the location of the cameras. We know that the optimal position of the camera was in front of the employee; however, putting them there could affect the behavior of the employees, and that is an undesirable situation. We installed the system on the computers that are already installed in the company without notifying the operator of the details of the experiment. On the other hand, we chose locations where only one employee was in front of the camera to improve the performance of the system since, although the camera could recognize up to three persons at the same time, this considerably reduced the response time, causing sampling errors. Finally, this work seeks to establish a reference between emotional states and cognitive ergonomics; up to our best knowledge, no model clearly relates them, so we present a proposal that adapts classic and novel models of computation and ergonomics that can be applied as an engineering solution.

**Author Contributions:** Conceptualization, A.R.A. and A.R.-V; methodology, A.R.A. and A.R.-V.; software, A.R.A; validation, A.R.A. and A.R.-V.; formal analysis, A.R.A..; investigation, A.R.A. and A.R.-V.; resources, A.R.A. and A.Q.; data curation, A.R.A.; writing—original draft preparation, A.R.A., A.R.-V and A.Q.; writing—review and editing, A.R.A. and A.Q.; visualization, A.R.A. and A.Q.; supervision, A.R.A.; project administration, A.R.A. and M.A.L.R.; funding acquisition, A.R.A., A.R.-V., M.A.L.R. and A.Q. All authors have read and agreed to the published version of the manuscript.

**Funding:** This research received no external funding.

**Acknowledgments:** Thanks to Tecnologico Nacional de México and Instituto Tecnologico de Tijuana.

**Conflicts of Interest:** The authors declare no conflict of interest.

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
