# Peer review of "Cognitive Ergonomics Evaluation Assisted by an Intelligent Emotion Recognition Technique"

_applsci, doi:10.3390/app10051736_

Round 1

Reviewer 1 Report

The main aim of the paper is to implement pattern recognition and cognitive analysis to identify facial expressions and correlates them to a situation awareness model, in order to measure the levels of comfort and mental stability of a worker in real-time.

The paper is not mature. I have many concerns about it. The main ones are:

1) the authors present the techniques they used but do not provide an adequate analysis of the state of the art. For instance, facial expression recognition is a very huge research field and the main approaches have to be listed and discussed before going deep into the adopted solution. Besides the reader is not able to understand the choice and the advantages with respect to other possible ones. 

2) experimental results are qualitative. how can we assess the approach in the considered application field? is there a ground truth? how many times the proposed approach failed?

Summing up the paper, in the present form, seems to be more suited for a conference or a workshop than for a journal. A great effort is necessary by authors to improve it before publication. 

Author Response

Description of the changes

  • A paragraph was added in section 1, which specifies why it is not feasible to use invasive features techniques to experiment (lines 43 to 50).
  • Sections 2.1.2.1 and 2.1.3 were annexed, explaining, and extending the theory about the recognition of emotions in the face and clarifying the structure of the neural network. (lines, 82 to 111).
  • The notations for Table 1 were fixed.
  • The table is annexed in section 2.3.3, which attempts to clarify the metadata of the information in the databases.
  • A confusión matrix was added in to Section 3.1. Figure 6- Line 260.

Reviewer 2 Report

The authors presented a novel solution to cognitive ergonomics evaluation of workers in the manufacturing industry. 

The methodology is well described and easy to follow.

However, the article lacks in literature review and discussion part. I can understand there is no direct comparison exists in the same setting but the author should write about other approaches in kind of similar/different settings such as using EEG caps/audio and visual information and discuss them in the discussion part. Why we cannot use audio or EEG for author's setting (or a possible future work). It is good to have a separate section for the 'conclusion and future work'. 

moreover, in section 2.3.3, it is better to show how many instances of each emotion the dataset contains. A histogram representation or Table.

Your emotion recognition system shown an accuracy of 88%, it is better to show the confusion matrix for it with precision and recall of each class.

In Table 1: What is meant by a single class, two class, three class.   

Some Reference (but the authors should explore more: Do a database search) 

Su, J. and Luz, S., 2016. Predicting cognitive load levels from speech data. In Recent Advances in Nonlinear Speech Processing (pp. 255-263). Springer, Cham.

Engström, Johan, Emma Johansson, and Joakim Östlund. "Effects of visual and cognitive load in real and simulated motorway driving." Transportation research part F: traffic psychology and behaviour 8.2 (2005): 97-120.

Anderson, Erik W., et al. "A user study of visualization effectiveness using EEG and cognitive load." Computer graphics forum. Vol. 30. No. 3. Oxford, UK: Blackwell Publishing Ltd, 2011.

Author Response

(The authors gave the same response as above.)

Round 2

Reviewer 1 Report

The improvements improved the paper even if I would have appreciated a deeper analysis of literature and a comparison with other approaches.
Summing up the paper could be accepted but there are some typos to be corrected and I suggest to further improve the overview of existing approaches by citing some survey paper about assistive technology such as
"Deep learning for assistive computer vision" in proc. of ECCV2018

"A brief review of facial emotion recognition based on visual information" in MDPI Sensors 2018
"Multimodal Emotion Recognition for Human-Computer Interaction: A Survey" in System 2017
and some strictly related paper such as
"Automatic emotion recognition in robot-children interaction for ASD treatment" in the Proceedings of the IEEE International Conference on Computer Vision 2015

Author Response

Thank you very much for the comments, in the introduction section a paragraph is attached in which the FER technique is explained in more detail, the recommended articles were taken into consideration, thank you very much for that.

Changes were also made in the writing of the document and some spelling mistakes were removed, such as lack of spaces and repeated words. We also worked on congruence in the writing of the document, as suggested.

Thank you.

Reviewer 2 Report

Thanks for the updated version. It looks way improved.

Some final comments.

there are some typos which need to be corrected such as no space in line 105 between text and references.

check line 237 

line 282

Figure 6: better to write the name of emotion instead of 1,2,3,4 

You should create a git repository of your system and make it available to the research community.

Authors are encouraged to demo their system at ICMI or INTERSPEECH

Author Response

Thank you very much for your comments, all the suggested changes were made and we also worked on the writing of the document. In addition, in Figure 6, you can read the legends by class, as suggested. On the other hand, thank you very much for your recommendations, we are working with our institution to release the code and to attend the suggested forums.

Thank you.